# The New Dipeptide TSPO Ligands: Design, Synthesis and Structure–Anxiolytic Activity Relationship [note 1]

**DOI:** 10.3390/molecules25215132

**Published:** 2020-11-04

**Authors:** Tatiana A. Gudasheva, Olga A. Deeva, Andrey S. Pantileev, Grigory V. Mokrov, Inna V. Rybina, Milada A. Yarkova, Sergei B. Seredenin

**Affiliations:** Federal State Budgetary Institution “Zakusov Research Institute of Pharmacology” (FSBI “Zakusov Institute of Pharmacology”), Baltiyskaya, 8, 125315 Moscow, Russia; olga.angstrem@gmail.com (O.A.D.); and.pantileev@academpharm.ru (A.S.P.); g.mokrov@gmail.com (G.V.M.); jarkova@academpharm.ru (I.V.R.); doctorpharm@mail.ru (M.A.Y.); seredeninpharm@mail.ru (S.B.S.)

**Keywords:** TSPO ligands, dipeptides, GD-102, molecular docking, anxiolytic activity, elevated plus maze test, open field test, PK11195

## Abstract

The translocator protein (TSPO, 18 kDa) plays an important role in the synthesis of neurosteroids by promoting the transport of cholesterol from the outer to the inner mitochondrial membrane, which is the rate-limiting step in neurosteroidogenesis. Stimulation of TSPO by appropriate ligands increases the level of neurosteroids. The present study describes the design, synthesis and investigation of anxiolytic-like effects of a series of *N*-acyl-tryptophanyl-containing dipeptides. These novel dipeptide TSPO ligands were designed with the original drug-based peptide design strategy using alpidem as non-peptide prototype. The anxiolytic activities were investigated in Balb/C mice using the illuminated open-field and elevated plus-maze tests in outbred laboratory mice ICR (CD-1). Dipeptide GD-102 (*N*-phenylpropionyl-l-tryptophanyl-l-leucine amide) in the dose range of 0.01–0.5 mg/kg intraperitoneally (i.p.) has a pronounced anxiolytic activity. The anxiolytic effect of GD-102 was abolished by PK11195, a specific TSPO antagonist. The structure–activity relationship study made it possible to identify a pharmacophore fragment for the dipeptide TSPO ligand. It was shown that l,d-diastereomer of GD-102 has no activity, and the d,l-isomer has less pronounced activity. The anxiolytic activity also disappears by replacing the C-amide group with the methyl ester, a free carboxyl group or methylamide. Consecutive replacement of each amino acid residue with glycine showed the importance of each of the amino acid residues in the structure of the ligand. The most active and technologically available compound GD-102, was selected for evaluation as a potential anxiolytic drug.

## 1. Introduction

TSPO (18 kDa; translocator protein) is currently considered as one of the most promising targets for developing new anxiolytic drugs without side effects of benzodiazepine tranquilizers. TSPO is a carrier protein located on the outer mitochondrial membrane. TSPO transports cholesterol from the outer to the inner mitochondrial membrane, which is a limiting step in the synthesis of neurosteroids, which positively modulate the function of the GABA_A_ receptor by their own binding sites other than benzodiazepine [1,2]

Over the past 25 years, about 20 different classes of TSPO ligands have been created, among which quite a few molecules have been found to have neuropsychoptropic activity in in vivo models [3,4]. An example of TSPO ligands is the compound CB-34 from the group of 2-phenylimidazo[1,2-a]pyridines, which exhibited a pronounced anti-conflict effect in the Vogel test in rats [5]; another is the indole derivative FGIN-1-27, which prevents the development of neophobia [6]; and another is compound YL-IPA08 from the group of pyridine derivatives, which exhibits antidepressant and anti-anxiety effects in the different animal models [7]. Mokrov et al. developed a series of the new pyrrolo[1,2-a]pyrazine TSPO ligands [8]. Among them, compound GML-1 (*N*-benzyl-*N*-methyl-1-phenylpyrrolo[1,2-a]pyrazine-3-carboxamide) possessed anxiolytic, antidepressant and nootropic activities and had no side effects of benzodiazepine tranquilizers [9] (Scheme 1).

Despite the fact that the well-known TSPO ligands have significant efficacy in various animal models of anxiety [10], depression [11] and in a number of other behavioral tests [3,4], none of them have passed clinical trials or is used in clinical practice. Perhaps that is mainly due to theirs side effects associated with the xenobiotic nature of these compounds.

An original strategy for the creation of highly active, non-toxic, physiological drugs based on dipeptides was developed in the Zakusov Research Institute of Pharmacology [12,13]. The strategy consists of constructing such dipeptides that are structurally similar to non-peptide biologically active compounds. This approach was called drug-based peptide design (DBPD). Due to their low molecular weight and the presence of an active transport system, dipeptides are able to penetrate biological barriers. A small number of bonds that are targets of proteases provide their enzymatic stability up to the possibility of oral administration. Finally, they are easy to synthesize and cheap to manufacture.

As an example, the dipeptide nootropic drug Noopept (*N*-phenylacetyl-l-prolylglycine ethyl ester) has been created using the presented strategy starting from the structure of the classical nootropic drug piracetam (*N*-carbamidomethylpyrrolidone-2) [14,15]. Noopept has been on the pharmaceutical market since 2006 (www.noopept.ru).

Using the DBPD approach the world’s first original dipeptide TSPO ligand *N*-carbobenzoxy-l-tryptophanyl-l-isoleucine amide (GD-23) was designed and synthesized in our laboratory. This compound demonstrated the presence of anxiolytic activity in the widely used pharmacological tests in rodents in the dose range of 0.1–1.0 mg/kg intraperitoneally (i.p.). The ligand properties of GD-23 were proved by pharmacological inhibitory analysis using a selective antagonist TSPO and two blockers of neurosteroidogenesis [16,17].

In this work we developed the further group of tryptophan-containing dipeptide TSPO ligands and studied the structure and anxiolytic activity relationship in this series.

## 2. Results and Discussion

### 2.1. Design and Molecular Docking

We used alpidem [18] as a non-peptide prototype for the design of the peptide TSPO ligand by DBPD method. Alpidem’s structure contains two aromatic nuclei (phenyl and imidazopyridine) and the branched aliphatic chain. These groups can be simulated with phenylalanine’s, tryptophan’s and isoleucine’s or leucine’s side chains, respectively. Furthermore, an amide group in alpidem’s structure is a possible prototype of a peptide bond (Figure 1). Thus, alpidem can be imitated by the following tripeptide sequences: –Phe–Trp–Ile– or –Phe–Trp–Leu–.

While designing TSPO dipeptide ligands based on these tripeptide sequences, we kept the tryptophan and isoleucine (leucine) residues unchanged in the sequence –Trp–Ile– (for GD-23) or –Trp–Leu– (for GD-102). The phenylalanine residue was replaced with the isomorphic bioisosteres-benzyloxycarbonyl (for GD-23) or phenylpropionyl (for GD-102) groups (see Scheme 2). These residues do not have a charged amino group like alpidem and the putative tripeptide sequences. The possible residue following after Ile (or Leu) was changed by amide group, whereby there was no negatively charged carboxyl group, as in alpidem. Since alpidem contains a symmetrical aliphatic fragment, we expected that *N*-phenylpropionyl-Trp-Leu–NH_2_ would be more active than benzyloxycarbonyl-Trp-Ile-NH_2_, which we described earlier [17].

To confirm that GD-102 can be TSPO ligand like GD-23 and alpidem, we performed their comparative molecular docking into TSPO active site. The spatial structure of the TSPO receptor in a complex with PK11195 (classical TSPO ligand), which was obtained by NMR (PDB ID: 2MGY), was used for docking studies [19]. Calculations were performed in the semi-rigid docking mode using the Glide software [20]. This software determines the complexes of ligand–receptor with the minimum glide gscore and glide emodel values (Table 1). PISA program was used for calculation the interface of TSPO and studying compounds [21,22]. The Swiss PDB Viewer program (https://spdbv.vital-it.ch) was used to reveal amino acids’ moieties that make contact with individual ligand fragments. The docking results were visualized using PyMol program (https://pymol.org/2/).

According to the docking data, the molecule of alpidem is located in the TSPO binding site in the following way. The bicyclic aromatic nucleus forms van der Waals contacts with the residues Ala23, Val26, Pro40, Pro45 and Arg46; the phenyl core generates contacts with Ala23, Trp95, Leu114 and Trp143. The aliphatic zone of the alpidem interacts with Ala23, Val26, Trp95, Trp107, Ala110, Ala147 and Leu150 (Figure 1 and Figure 2a).

GD-23 molecule is docked into the TSPO binding site in such a way that the tryptophan side group (indole fragment) is surrounded by Ala23, Leu49, Trp95, Leu114, Trp143 and Ala147 moieties. The phenyl ring of the benzyloxycarbonyl group is bound to Val26, Ser41, His43, Arg46 and Trp107. The isoleucine side chain (aliphatic fragment) is surrounded by Val26, Trp95, Trp107, Ala110, Leu114, Ala147 and Leu150 (Figure 1 and Figure 2b).

GD-102 side group of the tryptophan moiety forms contacts with Ala23, Val26, Leu31, Arg46 and Leu49. The phenyl ring of the *N*-acyl group forms contacts with Ala23, Ile52 and Leu114; the aliphatic part with Ala23, Val26, Ala110 and Leu150 (Figure 1 and Figure 2c). Moreover, the distribution of contact areas by the amino acid residues of the TSPO interface is almost the same for the all three compounds (alpidem, GD-23 and GD-102). The largest contact areas are in Ala23, Val26, Trp143 and Leu150 (Figure 1 and Figure 2c)

When we superimposed alpidem and GD-23 (Figure 3), their aliphatic fragments fit together. The phenyl nucleus of GD-23 *N*-benzyloxycarbonyl group overlaps with the bicyclic nucleus of alpidem. The planes of aromatic nuclei are located at an angle of 90 degrees to each other. The side group of the tryptophan moiety of GD-23 is overlaid with the phenyl nucleus of alpidem. The surroundings of GD-23 phenyl nucleus reproduce the surroundings of the alpidem bicyclic core (with the exceptions of Ala23 and Leu49 residues not being included in the surroundings of phenyl GD-23, and the remainder of Leu31, which is not surrounded by alpidem bicyclic core), and the surroundings of the tryptophan moiety of GD-23 almost do not differ from the surroundings of the phenyl group of alpidem, with the exceptions of residues of Leu49, Phe46 and Ala147, which are not included in the surroundings of the tryptophan moiety of GD-23.

At the same time, the superposition of the GD-102 with the alpidem (Figure 4) in their docked conformation leads to the combination of bicyclic and phenyl nuclei and aliphatic parts of the structures. GD-102 is located in the TSPO binding site close to the alpidem location. The largest difference concerns the bicyclic nuclei of the ligands located at an angle of 90 degrees to each other. The interfaces of alpidem and GD-102 are very similar both in composition and in the distribution of the contact surface over the amino acid residues of the receptor. The largest contact areas are present for Ala23, Val26, Leu150 and Trp143. The largest contact area with the GD-102 includes Trp143 of the TSPO tryptophan residues. The next most significant are Trp107 and Trp95.

GD-23 and GD-102 are superimposed in a way that the phenyl nuclei of these molecules are combined with indolyl (Figure 5). Only the side aliphatic groups of leucine and isoleucine are similarly located. The phenyl core of GD-23 is perpendicular to the indole core of GD-102. Such serious changes occurred when the benzyloxycarbonyl group was replaced by a phenylpropionyl group and the isoleucine moiety was replaced by the leucine moiety when we moved from GD-23 to GD-102.

Thus, docking data suggest that both dipeptide compounds are TSPO ligands. A similar posture of GD-102 to alpidem in the complex with the receptor may be the reason for its greater biological activity in comparison to GD-23.

### 2.2. Chemistry

The synthetic route for the preparation of the dipeptides is represented in the Scheme 3. The method of carboxyl group activation using succinimide esters by Anderson technique was used for the dipeptide GD-102 synthesis [23]. Boc–leucine was obtained at first. Its activated *N*-hydroxysuccinimide ester was obtained at the second stage, which further underwent the ammonolysis reaction by the solution of aqueous ammonia; the Boc–protecting group was removed by acidolysis in the presence of trifluoroacetic acid to give trifluoroacetate of the leucine amide. At the next stage, the carboxyl group of *Z*-l-tryptophan was activated by *N*-hydroxysuccinimide to obtain its activated ester; the unblocked leucine amide was acylated by the *Z*-l-tryptophan activated ester. The hydrogenolysis with palladium catalyst was used to remove the *Z*-protecting group of tryptophan. At the next stage, the activated *N*-hydroxysuccinimide ester of phenylpropionic acid was obtained. The unblocked l-tryptophanyl-l-leucine amide was acylated by the phenylpropionic acid activated ester to give the target *N*-phenylpropionil derivative of this dipeptide. The total yield from *Z*-tryptophan was 42% (see Scheme 3).

Diastereomeres of GD-102 (*N*-phenylpropionyl-d-Trp-l-Leu–NH_2_ GD-128 and *N*-phenylpropionyl-l-Trp-d-Leu–NH_2_ GD-123 respectively) were obtained to reveal the effect of the amino acid residues’ configuration on the anxiolytic activity. The corresponding methylamide (*N*-phenylpropionyl-l-Trp-l-Leu–NHCH_3_ GD-119), methyl ether (*N*-phenylpropionyl-l-Trp-l-Leu–OMe GD-116) and a compound with a free carboxyl group (*N*-phenylpropionyl-l-Trp-l-Leu–OH GD-118) were obtained to identify the effect of C-substitution nature on activity. To reveal the role of the side radicals of amino acids, two glycine analogues of the compound GD-102 were synthesized, in which each of the amino acid residues was successively replaced by a glycine residue (compounds Ph(CH_2_)_2_C(O)-Gly-l-LeuNH_2_ and Ph(CH_2_)_2_C(O)-l-Trp-GlyNH_2_, GD-129 and GD-125, respectively). To reveal the nature of *N*-substitution, an analogue with a shortened phenylalkanoyl substituent chain, PhCH_2_C(O)-l-Trp-l-LeuNH_2_ (GD-107), and a compound with an extended chain but not containing an aromatic nucleus, CH_3_(CH_2_)_4_C(O)-l-Trp-l-LeuNH_2_ (GD-108), were synthesized. GD-102 analogues were synthesized similarly to itself (see Scheme 3). The structures of the compounds were confirmed by NMR; the substances were characterized by such physicochemical parameters as thin layer chromatographic mobility, melting temperature and optical rotation angle.

### 2.3. Anxiolytic Activity

#### 2.3.1. Open Field Test

The anxiolytic activity of every synthesized dipeptide was studied in a modified open field (OF) test with a flash light [24] using male Balb/c mice with freezing reaction [25]. After the OF test, all substances were evaluated on spontaneous locomotor activity using an infrared actimeter Panlab (Spain). It was shown that there were no statistical significant differences between experimental groups and control groups in general motor activity and total distance travelled. Thus, it permits us to exclude the stimulant activity of dipeptides. It was found in the OF test that the compound GD-102 in the dose range of 0.01–0.05 mg/kg i.p. prevented a fearful reaction in Balb/c mice compared to the control group (see Table 2), which speaks well for the presence of anxiolytic activity. Indeed, as the docking data assumed (Table 1), GD-102 is more active than GD-23 in a behavioral test (0.1–1.0 mg/kg; see Table 2) [26].

#### 2.3.2. The Structure–Anxiolytic Activity Relationship

The replacement of l-leucine in the structure of GD-102 with d-leucine leads to the disappearance of anxiolytic activity (Table 2). The l,d-diastereomer (GD-123) has no activity in the dose range of 0.1–1.0 mg/kg. At the same time, replacing l-tryptophan with d-tryptophan leads only to a decrease of activity. The minimum effective dose for the d,l-diastereomer (GD-128) increases by 50 times compared to GD-102 (from 0.01 to 0.5 mg/kg; see Table 2). These data indicate the stereospecificity of the anxiolytic effect of GD-102 and the importance of the natural configuration of the residues for the activity manifestation, especially in the case of the leucine residue in the ligand structure. Obviously, the d-configuration of leucine creates steric hindrances for dipeptide binding. Sequential replacement of amino acid residues in GD-102 with glycine (removal of side chains) leads to an activity decrease: upon transition from GD-102 to both GD-125 (*N*-phenylpropionyl-l-Trp-Gly-NH_2_) and GD-129 (*N*-phenylpropionyl-Gly-l-Leu), the minimum effective dose is increased from 0.01 mg/kg to 0.5 mg/kg. This indicates the importance of the side groups of both amino acid residues of GD-102 for the manifestation of activity.

Replacement of the aryl-containing phenylpropionyl *N*-terminal substituent in GD-102 with the aliphatic *N*-hexanoyl substituent of equal length (compound GD-108) leads to a 50-times decrease in the activity at the minimum effective dose. At the same time, the shortening of the phenylpropionyl substituent to phenylacetyl (GD-107) leads only to a 10-times decrease in activity, which indicates the importance of the aromatic phenyl fragment in the *N*-acyl substituent.

The most active derivative at the C-terminus is GD-102 with an unsubstituted amide group. It exhibits activity at a dose of 0.01 mg/kg that is 10 times less than the minimum effective doses of compound with the free C-terminal group (GD-118) and the methyl ester (GD-116) or the methylamide (GD-119; see Table 2).

The analysis of the relationship between structure and activity in the investigated series of GD-102 analogues clarified the pharmacophore units of the peptide TSPO ligand. A dipeptide core and the aromatic nucleus at the distance of seven sigma-bonds from the indolyl fragment of the first residue are required for manifestation of activity. The indolyl fragment itself is important, but does not play a key role in the manifestation of anxiolytic activity since its replacement with the glycine moiety does not lead to a complete loss of activity. On the other hand, the leucine moiety is the ideal as the second residue of the dipeptide structure of ligand. The leucine replacement with glycine leads to a 50-times decrease of activity, and its replacement with D-leucine leads to a complete loss of activity. The phenyl fragment in the *N*-terminal substituent is required for activity, which indicates the possible presence of a phenylalanine residue in the putative endogenous TSPO peptide ligand. This is proved by the optimal length of the *N*-terminal phenylpropionyl substituent corresponding to the length of the phenylalanine residue.

It is interesting to note that the study of the anxiolytic activity of dipeptides Leu–Trp showed the absence of activity or the reversed effect [27].

The activity of a dipeptide with the natural l-configuration of its residues supports its similarity to the putative endogenous peptide TSPO ligand, whose structure can be represented as X–Phe–Trp–Leu–NH_2_. Further investigations are required for the final solution to this problem.

#### 2.3.3. Elevated Plus-Maze Test

The anxiolytic activity of the most active dipeptide GD-102 was also confirmed in the elevated plus-maze (EPM) test in outbred laboratory mice ICR (CD-1) (Appendix A). The EPM test is considered as the most adequate for anxiolytic activity detecting, which is expressed by increasing of the percentage of the animal’s residence time in the open arms and the relative frequency of entry into the open arms [28]. Dipeptide GD-102 in the dose range 0.1–0.5 mg/kg intraperitoneally demonstrated a significant (*p* < 0.05) anxiolytic effect in the relative time spent in the open arms. At the dose of 0.5 mg/kg GD-102 showed significant activity in both the relative time spent in the open arms and in the relative number of entries into the open arms of the labyrinth (Figure 6).

To prove the TSPO ligand properties of GD-102, we studied the effect of the classical TSPO antagonist-compound PK11195 (1(2-chlorophenyl)-*N*-methyl-(1-methylpropyl)-3-isoquinoline-carboxamide) in the EPM test. PK11195 inhibitor was administered intraperitoneally 30 min before GD-102, at the dose of 10.0 mg/kg. This dose of PK11195 had no effect on the time spent by animals in the open arms and on the frequency of entries to the open arms [29]. In each study, the animals received the same number of injections. The control animals received water.

Preliminary administration of PK11195 completely blocked the anxiolytic effect of GD-102 (Figure 7). Behavioral parameters of the animals in this group remained at the control level but significantly differed from the behavioral parameters of the mice from the group that received GD-102. The experimental results demonstrate the dependence of the anxiolytic effect of GD-102 on interaction with the TSPO binding site.

## 3. Experimental

### 3.1. Chemistry

The chemicals and amino acids reported here were obtained from Sigma-Aldrich (Munich, Germany) or Alfa Aesar (Tewksbury, MA, USA). The solvents were obtained from Himmed (Moscow, Russia). The solvents were purified and dried by standard methods. Thin-layer chromatography (TLC) using silica gel on aluminum sheets (Merck, Darmstadt, Germany) was used to observe the chemical reactions; chloroform and methanol (9:1) were used as the eluent. The TLC was visualized in ultraviolet light or using the tolidine test. The melting points were measured using Optimelt MPA 100 (Stanford Research Systems, Sunnyvale, CA, USA). ^1^H-NMR spectra were recorded using Bruker Fourier 300 (Bruker Corporation, Leipzig, Germany) with DMSO-*d*_6_ (solvent). Chemical shifts (δ) are indicated in ppm (part per million), with tetramethylsilane (TMS) as a reference; coupling constants (*J*) were recorded (Hz). The NMR shifts were documented as follows: s, singlet; d, doublet; and m, multiplet. The optical rotation values were measured using an ADP 410 automatic digital polarimeter (Bellingham Stanley, Tunbridge Wells, Kent, Great Britain) at the wavelength of the D-line of the sodium spectrum (589.3 nm). The values of specific optical rotations were calculated by the formula: [α]_D_ = (α × V) ÷ (l × m), where—α is observed optical rotation in degrees; V is the volume of solution in mL; l is the length of the measuring cell in dm; m is the weight of substance in grams.

*General procedure for the synthesis of N**-hydroxysuccinimide esters of carboxylic and amino acids***.** The mixture of 30.00 mmol acid in 200–300 mL of ethylacetate and 34.50 mmol (15% excess) of *N*-hydroxysuccinimide was cooled to +5 °C using ice-water bath. Then 35.10 mmol (17% excess) of dicylohexylcarbodiimide was added. The mixture was stirred for 30 min at +5 °C and then for 5 h at room temperature. The reaction was monitored by TLC. The dicyclohexylurea precipitate was filtered off and the solvent was removed in vacuo. The resulting viscous oil was dissolved in 100 mL of ethylacetate. The solution was kept for 24 h at +8 °C; the re-formed dicyclohexylurea precipitate was filtered off again; the filtrate was washed with 5% NaHCO_3_ (2 × 100 mL) and 100 mL of distilled water. The organics were dried over anhydrous Na_2_SO_4_ and then filtered; the solvent was removed in vacuo.

*N-Boc–l-Leu–OSu.* Glassy crystals; yield: 73%; m.p. 110–112 °C; ${\left[\alpha \right]}_{26}^{D}$ = −40° (c = 2, dioxane). (Lit. data: m.p. 116 °C (diisopropyl esther), ${\left[\alpha \right]}_{25}^{D}$ = −41.8° (c = 2, dioxane) [23]); ^1^H-NMR (DMSO-*d*_6_) δ, ppm: 0.82–0.88 (m, 6H, 2C^δ^H_3_ Leu), 1.39 (s, 9H, -OC(CH_3_)_3_), 1.45 (m, 2H, C^β^H Leu), 1.56–1.73 (m, 1H, C^γ^H_2_ Leu), 2.82 (m, 4H, OSu), 3.86–3.88 (m, 1H, C^α^H Leu), 7.93 (d, *J* = 8.38 Hz, 1H, NH Leu). 

*Boc–d–Ile–OSu.* Oil (yield: 91%); *R*_f_ 0.90. ^1^H-NMR (DMSO-*d*_6_, δ, ppm): 0.86 (m, C^δ^H_3_ Ile, 3H), 0.97 (d, J = 6.7 Hz, C^γº^H_3_ Ile, 3H), 1.27 and 1.52 (two m, C^γ^H_2_ Ile, 2H), 1.40 (s, -OC(CH_3_)_3_, 9H), 1,87 (1H, m, C^β^H Ile), 2,80 (4 H, m, OSu), 4.24 (m, C^α^H Ile, 1H), 7.57 (1 H, d, *J* = 8.40 Hz, NH Ile).

*N-Z-l-Trp–OSu.* Cream foam (yield: 92%), m.p. 137–140 °C ${\left[\alpha \right]}_{25}^{D}$ = −60.0° (c = 1, DMF); *R*_f_ 88. ^1^H-NMR (DMSO-*d*_6_) δ, ppm: 2.78 (m, 4H, OSu), 3.01–3.25 (m, 2H, C^β^H Trp), 3.98 (m, 1H, C^α^H Trp), 4.97 (s, 2H, -OCH_2_C_6_H_5_), 6.73–7.62 (m, 10H, Ar), 8.56 (d, *J* = 8.8 Hz 1H, NH Trp), 10.78 (1H, c, NH_indole_). 

*N-Z-d-Trp–OSu.* Cream foam (yield: 96%); ${\left[\alpha \right]}_{25}^{D}$ = +44.3 ° (c 1, DMF); *R*_f_ 87; ^1^H-NMR (DMSO-*d*_6_, δ, ppm): 2.80 (m, OSu, 4H), 3.07–3.25 (m, C^β^H Trp, 2H), 4.00 (m, C^α^H Trp, 1H), 4.95 (s, -OCH_2_C_6_H_5_, 2H), 6.94–7.77 (m, -OCH_2_C_6_H_5_, indole, 10H), 8.56 (d, *J* = 8.8 Hz, NH Trp, 1H), 10.80 (1H, s, NH indole).

*Boc–Gly–OSu.* White powder (yield 78%), m.p. 165–167 °C, *R*_f_ 0,66 (CHCl_3_/MeOH 9/1). ^1^H-NMR (DMSO-*d*_6_) δ, ppm: 1.39 (s, -OC(CH_3_)_3_, 9H), 2.81 (m, OSu, 4H), 4.08 (d, *J* = 6.1 Hz, CH_2_ Gly, 2H), 7.48 (t, *J* = 5.9 Hz, NH Gly, 1H).

*Ph(CH_2_)_2_C(O)–Gly–OSu.* Viscous white oil (yield 75%). ^1^H-NMR (DMSO-*d*_6_) δ, ppm: 2,37 (t, 2H, CH_2_CO), 2,69 (m, 2H, CH_2_C_6_H_5_), 2.80 (m, 4H, OSu), 3.56 (m, 2H, CH_2_ Gly), 7.27–7.43 (m, 5H, C_6_H_5_,), 7.47 (t, *J* = 5.8 Hz, NH Gly, 1H).

*Ph(CH_2_)_2_C(O)Su.* White crystals (yield: 45%) m.p. 83–84 °C. ^1^H-NMR (DMSO-*d*_6_) δ, ppm: 2.81 (m, 4H, OSu), 2.97–3.01 (m, 2H, CH_2_C_6_H_5_), 3.34 (s, 2H, CH_2_CO), 7.16–7.31 (m, 5H, Ar).

*PhCH_2_C(O)–OSu.* White powder (yield 96%); m.p. 111–112 °C. ^1^H-NMR (DMSO-*d*_6_) δ, ppm: 2.81 (m, OSu, 4H), 4.10 (s, CH_2_C_6_H_5_,2H), 7.20–7.42 (m, C_6_H_5_, 5H).

*CH_3_(CH_2_)_4_C(O)–OSu.* White glassy crystalls (yield 76%); m.p. 53–55 °C; *R*_f_ 0.85; ^1^H-NMR (DMSO-*d*_6_) δ, ppm: 0.87 (m, 3H, CH_3_), 1.32 (m, 2H, C^δ^H_2_,), 1.62 (m, 4H, C^γ^H_2_C^β^H_2_), 2.63–2.68 (m, 2H, C^α^H_2_), 2,80 (m, OSu, 4H).

### 3.2. Preparation of Starting Compounds

*Boc–Gly–NH_2_.* To a solution of 6.00 g (27 mmol) Boc–Gly–OSu in 20 mL of DMF, 40 mL of aqueous ammonia was added. The resulting suspension was sustained at room temperature for 24 h without stirring. The resulting solution was concentrated by rotary evaporation. Yellow oil was obtained in the yield of 99%. (lit data m.p. 91–92 °C [30]) ^1^H-NMR (DMSO-*d*_6_) δ, ppm: 1.38 (s, -OC(CH_3_)_3_, 9H), 4.10 (d, *J* = 6.1 Hz, CH_2_ Gly, 2H), 7.49 (t, *J* = 5.9 Hz, NH Gly, 1H), 7.41 and 7.10 (two s, NH_2_ amide, 2H).

*N-Boc–l-Leu–NH_2_.* To a solution of 8.00 g (24.36 mmol) of Boc–l-Leu–OSu in 15 mL of DMF, 100 mL of an aqueous solution of ammonia was added. The resulting suspension was stirred for 30 min. The precipitate was kept for 3 h at +8°. The product was filtered, washed with distilled water until neutral and dried at 100 °C. The yield was 3.37 g (60%) of Boc–l-Leu–NH_2_ as white powder with m.p. 164–165 °C, ${\left[\alpha \right]}_{26}^{D}$ = −11.0° (c = 1, DMF). (Lit. data: m.p. 150–152 C, ${\left[\alpha \right]}_{24}^{D}$ = −11.9° (c = 1, methanol) [31]) ^1^H-NMR (DMSO-*d*_6_) δ, ppm: 0.82–0.85 (m, 6H, 2C^δ^H_3_ Leu), 1.08–1.22 (m, 1H, C^γ^H_2_ Leu), 1.36 (s, 9H, -OC(CH_3_)_3_), 1.45 (m, 2H, C^β^H Leu), 3.86–3.88 (m, 1H C^α^H Leu), 6.72 (d, *J* = 8.2 Hz, 1H, NH Leu), 6.89 and 7.19 (2s, 2H, NH_2_amide).

*N-Boc–d-Leu–NH_2_.* Obtained similarly to Boc–l-Ile–NH_2_ using Boc–d-Leu–OSu. White powder; yield 75%; m.p. 148–149 °C; ${\left[\alpha \right]}_{25}^{D}$ = +88.4° (c = 1, DMF); *R*_f_ 0.47 (CHCl_3_:MeOH 9:1). ^1^H-NMR (DMSO-*d*_6_) δ, ppm: 0.82–0.87 (m, 6H, 2C^δ^H_3_ Leu), 1.08–1.22 (m, 1H, C^γ^H Leu), 1.37 (s, 9H, -OC(CH_3_)_3_), 1.45 (m, 2H, C^β^H_2_ Leu), 3.86–3.89 (m, 1H C^α^H Leu), 6.72 (d, *J* = 8.38 Hz, 1H, NH Leu), 6.88 and 7.03 (2s, 2H, NH_2_amide).

*TFA*[l-Leu–NH_2_].* The suspension of 12.00 g (52.10 mmol) of Boc–l-Leu–NH_2_ in 30 mL of dichloromethane was treated with 60 mL of TFA (150% excess) and stirred for 2 h at room temperature. The solution was re-evaporated with diethyl ether (3 × 50 mL). The obtained white oil was triturated with diethyl ether, a white powder was obtained and the ether was decanted. The product was dried on air at room temperature. The yield was 11.88 g (99%) of TFA*[H-Leu–NH_2_], white powder with m.p. 118–119 °C, ${\left[\alpha \right]}_{25}^{D}$ = −18° (c = 1, DMF). ^1^H-NMR (DMSO-*d*_6_) δ, ppm: 0.83–0.89 (m, 6H, 2C^δ^H_3_ Leu), 1.08–1.22 (m, 1H, C^γ^H Leu), 1.45 (m, 2H, C^β^H_2_ Leu), 3.69 (m, 1H, C^α^H Leu), 7.52 and 7.93 (2s, 2H, NH_2_ amide), 8.12 (broad s, 3H, *N*^+^H_3_ Leu).

*TFA*[d-Leu–NH_2_].* Obtained similarly to TFA*[l-Leu–NH_2_] Boc–d-Leu–NH_2_. Yield 99%; white powder; m.p. 134–135; ${\left[\alpha \right]}_{25}^{D}$= +34.4°, (c = 1, DMF); ^1^H-NMR (DMSO-*d*_6_) δ, ppm: 0.88–0.90(6 H, m, 2C^δ^H_3_ Leu), 1.08–1.27 (1 H, m, C^γ^H Leu), 1.55 (2 H, m, C^β^H_2_ Leu), 3.63 (1H, m, C^α^H Leu), 7.53 and 7.90 (2H, 2s, NH_2_ amide), 8.07 (3H, broad s, *N*^+^H_3_ Leu).

*Z-l-Trp-l-Leu–NH_2_.* 3.30 g (13.52 mmol) of TFA*[l-LeuNH_2_] was dissolved in 30 mL of DMF with the addition of 2.35 mL (13.52 mmol) of DIPEA. This mixture was stirred for half an hour. Then 7.06 g (16.22 mmol, 20% excess) of *Z*-l-TrpOSu in 30 mL of DMF was added. The reaction mixture was stirred for 12 h at room temperature. The solvent was evaporated. The resulting fluent orange oil was dissolved in 200 mL of ethylacetate; washed with 3% H_2_SO_4_ (2 × 100 mL), 5% NaHCO_3_ solution (2 × 100 mL) and distilled water (1 × 100 mL); the organics were dried over Na_2_SO_4_; the solvent was evaporated. The resulting oil was dissolved in a minimum amount of DMF and diluted with 200 mL of distilled water; the white precipitate was obtained. The precipitate was maintained for 12 h at +5 °C, filtered, and washed with distilled water and hexane. The product was dried in vacuo over CaCl_2_ and paraffin. The yield was 5.78 g (94%), white powder with m.p. 166–169 °C; ${\left[\alpha \right]}_{26}^{D}$ = −31° (c = 1, DMF). ^1^H-NMR (DMSO-*d*_6_) δ, ppm: 0.82–0.88 (m, 6H, 2C^δ^H_3_ Leu), 1.08–1.22 (m, 1H, C^γ^H_2_ Leu), 1.45 (m, 2H, C^β^H_2_ Leu), 2.90–3.09 (m, 2H, C^β^H Trp), 4.28–4.31 (m, 2H, C^α^H Leu and C^α^H Trp), 4.93 (m, 2H, CH_2_CO), 6.99–7.30 (m, 10H, Aryl), 7.25 and 7.33 (2s, 2H, NH_2_ amide), 7.62 (d, *J* = 7.53 Hz, 1H, NH Leu), 7.97 (d, *J* = 7.83 Hz, 1H, NH Trp), 10.81 (s, 1H, NH indole).

*Z-d-Trp–l-Leu–NH_2_* Obtained similarly to Z-l-Trp-l-Leu–NH_2_ using Z-d-Trp–OSu. White powder (yield 73%); m.p. 180–122 °C, [α]^D^_23_ = +28.0° (c 1, DMF). ^1^H-NMR (DMSO-*d*_6_) δ, ppm: 0.71–0.80 (6H, m, 2C^δ^H_3_ Leu), 1.22–1.27 (1H, m, C^γ^H Leu), 1.40–1.45 (2H, m, C^β^H_2_ Leu), 2.93–3.04 (2H, 2m, C^β^H Trp), 4.16 (1H, m, C^α^H Leu), 4.31 (1H, m, C^α^H Trp), 4.96 (2H, m, -OCH_2_C_6_H_5_), 6.97–7.32 (10H, m, Aryl), 7.15 and 7.32 (2H, 2s, NH_2_ amide), 7.62 (1 H, d, *J* = 8.75 Hz, NH Trp), 7.97 (1H, d, *J* = 7.73 Hz, NH Leu), 10.81 (1H, s, NH indole).

*Z-l-Trp–d-Leu–NH_2_* was obtained similarly to Z-l-Trp-l-Leu–NH_2_ using TFA*[d-Leu–NH_2_]. White powder (yield 76%); m.p. 188–189 °C. ^1^H-NMR (DMSO-*d*_6_) δ, ppm: 0.82–0.88 (6 H, m, 2C^δ^H_3_ Leu), 1.45 (2H, t, C^β^H_2_ Leu), 1.56–1.73 (1H, m, C^γ^H Leu), 2.90–3.09 (2H, m, C^β^H Trp), 4.22 (1H, m, C^α^H Leu), 4.54 (1H, m, C^α^H Trp), 6.99–7.20 (10H, m, Aryl), 7.33 and 7.60 (2H, 2 s, NH_2_ amide), 7.93 (1H, d, *J* = 8.47 Hz, NH Leu), 8.08 (1H, d, *J* = 7.73 Hz NH Trp), 10.78 (1H, s, NH indole).

*H-l-Trp–l-Leu–NH_2_.* Through the suspension of 5.78 g (12.8 mmol) of *Z*-l-Trp-l-Leu–NH_2_ and 0.500 g of 10% Pd/C in 50 mL of methanol a stream of hydrogen was passed from the gas-holder for 2 h at vigorous stirring. After reaction was complete, the catalyst was filtered and washed with methanol. The methanol was evaporated in vacuo. The product was obtained in the amount of 4.71 g (99%) as a gray foam without a clear melting point, ${\left[\alpha \right]}_{26}^{D}$ = −25° (c = 1, DMF). ^1^H-NMR (DMSO-*d*_6_) δ, ppm: 0.82–0.88 (2dd, *J* = 12.95 Hz and *J* = 12.85 Hz, 6H, 2C^δ^H_3_ Leu), 1.09–1.26 (m, 1H, C^γ^H_2_ Leu), 1.45 (m, 2H, C^β^H_2_ Leu), 2.90 and 3.08 (2dd, 2H, C^β^H Trp), 3.50 (m, 2H, C^α^H Trp), 4.14 (m, 1H, C^α^H Leu), 6.94–7.55 (m, 5H, Ar), 7.05 and 7.41 (2s, 2H, NH_2_ amide), 7.60 (d, *J* = 7.83 Hz 1H, NH Leu), 8.05 (d, 1H, *J* = 8.29 Hz NH Trp), 10.85 (broad s, 1H, NH indole).

*H-l-Trp–d-Leu–NH_2_.* Obtained similarly to H-l-Trp-l-Leu–NH_2_ using *Z*-l-Trp-d-Leu–NH_2_. Grey powder (yield 96%); m.p. 110 °C. ${\left[\alpha \right]}_{26}^{D}$ = −5° (c = 1, DMF)^1^H-NMR (DMSO-*d*_6_) δ, ppm: 0.78–0.83 (m, 6H, 2C^δ^H_3_ Leu), 0.88 (m, 2H, C^γ^H_2_ Leu), 1.40 (m, 1H, C^β^H Leu), 2.71−3.03 (m, 2H, C^β^H_2_ Trp), 3.50 (m, 2H, C^α^H Trp), 4.21 (dd, *J* = 14.99 Hz and *J* = 7.26 Hz, 1H, C^α^H Leu), 6.96–7.33 (m, 5H, Ar), 7.05 and 7.41 (2s, 2H, NH_2_ amide), 7.55 (d, *J* = 7.92 Hz, 1H, NH Leu), 7.94 (d, *J* = 7.93 Hz, 1H, NH Trp), 10.82 (broad s, 1H, NH indole). 

*H-d-Trp–l-Leu–NH_2_.* Obtained similarly to H-l-Trp-l-Leu–NH_2_ using *Z*-d-Trp-l-Leu–NH_2_. Cream foam without a clear melting point (yield 99%), ${\left[\alpha \right]}_{26}^{D}$ = −17° (c = 1, DMF). ^1^H-NMR (DMSO-*d*_6_) δ, ppm: 0.81–0.88 (2dd, *J* = 12.95 Hz and *J* = 12.85 Hz, 6H, 2C^δ^H_3_ Leu), 0.88 (m, 2H, C^γ^H_2_ Leu), 1.40 (m, 1H, C^β^H Leu), 2.71–3.03 (m, 2H, C^β^H_2_ Trp), 3.50 (m, 2H, C^α^H Trp), 4.21 (m, 1H, C^α^H Leu), 6.96–7.33 (m, 5H, Ar), 7.05 and 7.41 (2s, 2H, NH_2_ amide), 7.55 (d, *J* = 7.54 Hz, 1H, NH Leu), 7.94 (d, *J* = 7.64 Hz, 1H, NH Trp), 10.82 (broad s, 1H, NH indole).

*Z-l-Trp–l-Leu–OMe.* In total, 2.43 g (13.36 mmol) of HCl*[l-Leu–OMe] was dissolved in 30 mL of DMF with 2.54 mL (14.57 mmol) of DIPEA; this mixture was stirred for half an hour. Then 6.98 g (16.03 mmol) of Z-l-Trp–OSu was added. The mixture was stirred for 24 h at room temperature. The solvent was evaporated; the resulting fluid oil was diluted with 300 mL of ethylacetate. The solution was washed with 3% H_2_SO_4_ (2 × 300 mL), and then with 5% NaHCO_3_ (2 × 300 mL) and distilled water (1 × 300 mL). The organic fraction was dried over anhydrous MgSO_4_ for half an hour, and then filtered and evaporated to a state of fluid oil. This oil was diluted with 200 mL of distilled water. The white precipitate was obtained, the water was decanted and the product was dried in vacuo over CaCl_2_ and paraffin. The yield was 5.95 g (95.6%) of the thick white oil without a clear melting point, *R*_f_=0.79, ${\left[\alpha \right]}_{26}^{D}$ = −21.8° (c = 1, DMF). (Lit data: m.p. 55–57 °C [32]). ^1^H-NMR (DMSO-*d*_6_) δ, ppm: 0.83–0.88 (m, 6H, 2C^δ^H_3_ Leu), 1.17 (m, 2H, C^γ^H Leu), 1.25 (m, 1H, C^β^H_2_ Leu), 2.96–3.10 (m, 2H, C^β^H Trp), 3.62 (s, 3H, -OCH_3_), 4.29 (m, 1H, C^α^H Leu), 4.41 (m, 1H, C^α^H Trp), 4.96 (m, 2H, -OCH_2_C_6_H_5_), 6.98–7.51 (m, 10H, Aryl), 7.66 (d, *J* = 7.64 Hz, 1H, NH Trp), 8.27 (d, *J* = 7.54 Hz, 1H, NH Leu), 10.83 (s, 1H, NH indole).

*H-l-Trp–l-Leu–OMe.* Through the suspension of 5.79 g (12.44 mmol) of *Z*-l-Trp-l-Leu–OMe and 1.00 g of 10% Pd/C in 100 mL of methanol the stream of hydrogen was bubbled for 2 h with vigorous stirring. After the reaction was complete the catalyst was filtered and washed with methanol. Methanol was removed in vacuo; 4.15 g of product was obtained as grey thick oil with ${\left[\alpha \right]}_{26}^{D}$ = −28.6° (c = 1, DMF). ^1^H-NMR (DMSO-*d*_6_) δ, ppm: 0.83–0.88 (m, 6H, 2C^δ^H_3_ Leu), 1.06–1.35 (m, 2H, C^β^H_2_ Leu), 1.58 (m,1H, C^γ^H Leu), 2.90–3.06 (m, 2H, C^β^H_2_ Trp), 3.63 (s, 3H, -OCH_3_), 4.29 (dd, 1H, C^α^H Leu), 4,41 (m, 1H, C^α^H Trp), 6.98–7.51 (m, 1H, Aryl), 7.67 (1d, *J* = 7.92 Hz, 1H, NH Trp), 8.27 (d, *J* = 7.92 Hz, 1H, NH Leu), 10.83 (s, 1H, NH indole).

### 3.3. Preparation of Target Dipeptides

*N-Ph(CH_2_)_2_C(O)–l-Trp–l-Leu–OMe (GD-116).* 3.67 g (14.85 mmol, 15% excess) phenylpropionic acid succinimide ester was added to the solution of 4.10 g (12.37 mmol) of H-l-Trp–l-Leu–OMe in 70 mL DMF. The solution was stirred for 12 h. Excess of the succinimide ester was quenched by adding 0.2 mL of DMAPA. The solvent was evaporated; diluted with 300 mL of ethylacetate; and washed with 3% H_2_SO_4_ (1 × 300 mL), 5% NaHCO_3_ (2 × 300 mL), and a saturated solution of NaCl (1 × 300 mL), and then with distilled water (1 × 300 mL). The organic fraction was dried over anhydrous MgSO_4_, evaporated to dryness. The product in a form of white oil was twice re-evaporated with diethyl ether to obtain stable white foam. The foam was crushed into powder and washed with hexane on the glass filter. The product was dried in vacuo over Na_2_SO_4_ and paraffin. The yield was 4.25 g (74%) of white powder with m.p. 134–135 °C, ${\left[\alpha \right]}_{26}^{D}$ = −41.4° (c = 1, DMF). ^1^H-NMR (DMSO-*d*_6_) δ, ppm: 0.84–0.91 (6 H, m, 2C^δ^H_3_ Leu), 1.06–1.27 (2H, m, C^β^H Leu), 1.48–1.58 (1H, m, C^γ^H_2_ Leu), 2.32 (2H, m, CH_2_ chain), 2.50 (H, m, CH_2_ chain), 2.92–3.09 (2H, m, C^β^H Trp), 3.61 (3H, s, -OCH_3_), 4.32 (1H, m, C^α^H Leu), 4,63 (1H, m, C^α^H Trp), 6.93–7.61 (10H, m, Aryl), 8.02 (1H, d, *J* = 7.73 Hz, NH Trp), 8.35 (1 H, d, *J* = 7.29 Hz, NH Leu), 10.91 (1 H, s, NH indole). 

*N-Ph(CH_2_)_2_C(O)-l-Trp-l-Leu–OH (GD-118).* The solution of 0.19 g (4.64 mmol, 200% excess) NaOH in 10 mL of water was added dropwise to the solution of 1.10 g (2.32 mmol) of *N*-Ph(CH_2_)_2_C(O)-l-Trp-l-Leu–OMe in 20 mL of methanol. The mixture was stirred for 3 h, and then diluted with 100 mL of cold water (+5–8 °C); finally, it was acidified with 5% H_2_SO_4_ to pH 4–5. The product was obtained by extraction with ethyl acetate (3 × 30 mL). Organics were dried over Na_2_SO_4_, the solvent was removed in vacuo, and then re-evaporated 3 times with diethyl ether to obtain the white foam. The foam was crushed, put to a glass filter and washed with hexane. The yield was 0.95 g (91%) with m.p. 91–92 °C, ${\left[\alpha \right]}_{29}^{D}$ = −19.8° (c = 1, DMF). ^1^H-NMR (DMSO-*d*_6_) δ, ppm: 0.83–0.88 (6H, m, 2C^δ^H_3_ Leu), 1.07–1.20 (2H, m, C^β^H Leu), 1.58 (1H, m, C^γ^H_2_ Leu), 2.37 (2H, m, CH_2_chain), 2.68 (H, m, CH_2_chain), 2.79–2.88 (2H, m, C^β^H Trp), 3.61 (3 H, s, -OCH_3_), 4.23–4.30 (1H, m, C^α^H Leu), 4.59–4.65 (1H, m, C^α^H Trp), 6.95–7.31 (10H, m, Aryl), 7.96 (1H, d, *J* = 8.29 Hz, NH Trp), 8.35 (1H, d, *J* = 8.01 Hz, NH Leu), 10.91 (1 H, s, NH indole) 12.63 (1H, broad s, OH).

*N-Ph(CH_2_)_2_C(O)-l-Trp-Leu–NHCH_3_ (GD-119).* In total, 1.10 g (2.37 mmol) of *N*-Ph(CH_2_)_2_C(O)–l-Trp–l-Leu–OMe was added to the fresh-made solution of methylamine in ethyl alcohol (23 g of solution containing 5.7 g of pure methylamine). The mixture was left in the closed flask for 6 days at room temperature without stirring. The solution was evaporated, and re-evaporated twice with diethyl ether to obtain white powder. The yield was 1.05 g (96%) of product with m.p. 201 °C (with decomposition), [α]${\left[\alpha \right]}_{29}^{D}$ = −16.6° (c = 1, DMF). ^1^H-NMR (DMSO-*d*_6_) δ, ppm: 0.81–0.88 (m 2C^δ^H_3_ Leu, 6H), 1.06–1.27 (m, C^β^H Leu, 2H), 1.58 (m C^γ^H_2_ Leu, 1H), 2.15 (s, -NHCH_3_, 3H), 2.37 (m, 2H CH_2_ chain), 2.69 (m, 2H chain), 2.87–3.95 (m, C^β^H Trp, 2H), 4.12 (m, C^α^H Leu, 1H), 4.35 (m, C^α^H Trp, 1H), 6.91–7.28 (m, 10H, Aryl), 7.60 (d, *J* = 8.20 Hz, NH Trp, 1H), 7.85 (d, *J* = 7.82 Hz, NH Leu, 1H), 10.83 (s, NH indole, 1H).

*N-Ph(CH_2_)_2_C(O)-l-Trp-l-Leu–NH_2_ (GD-102).* A total of 4.50 g (14.3 mmol) of H-l-Trp-l-Leu–NH_2_ was dissolved in 50 mL of DMF; then 3.86 g (15.6 mmol, 10% excess) of Ph(CH_2_)_2_C(O)Su was added and stirred for 12 h at room temperature. The solvent was evaporated to half the volume. The residue was diluted with 100 mL of water to obtain an insoluble product. The resultant crystals were washed with the glass filter with 100 mL of warm (43–45 °C) water, then with 100 mL of hexane and then with 50 mL of diethyl ether. The yield was 4.49 g (70%) of the product in the form of a white powder, with a m.p. 197–198 °C, ${\left[\alpha \right]}_{26}^{D}$ = −27° (c = 1, DMF). ^1^H-NMR (DMSO-*d*_6_, δ, ppm): 0.82–0,88 (6H, 2 dd, *J* = 12.95 Hz and J = 12.85 Hz, 2C^δ^H_3_ Leu), 1.45 (2H, t, C^β^H Leu), 1.56–1.73 (1 H, m, C^γ^H_2_ Leu), 2.38 (2H, m, CH_2_C_6_H_5_ chain), 2.69 (2H, m, CH_2_CO chain), 2.90–3.09 (2 H, 2dd, *J* = 9.66 Hz and *J* = 4.66 Hz, C^β^H Trp), 4.22 (1H, dd, *J* = 15.95 Hz and *J* = 7.64 Hz, C^α^H Leu), 4.54 (1H, m, C^α^H Trp), 6.99–7.20 (10H, m, Aryl), 7.33 and 7.60 (2H, 2 s, NH_2_ amide), 7.93 (1H, d, *J* = 8.29 Hz, NH Leu), 8.08 (1H, d, *J* = 8.01 Hz NH Trp), 10.80 (1 H, s, NH indole).

*N-PhCH_2_C(O)-l-Trp-l–Leu–NH_2_ (GD-108).* The 0.355g (1.76 mmol) of H-l-Trp-l-Ile-NH_2_ was dissolved in 20 mL of DMF; then 0.46g (2.11 mmol, 20% excess) of PhCH_2_C(O)Su was added and stirred for 12 h at room temperature. The solvent was evaporated to half the volume. The residue was diluted with 100 mL of water to obtain an insoluble product. The resulting precipitate was maintained for 12 h at room temperature, filtered over glassy filter and washed with 50 mL of water and 50 mL of hexane. The yield was 80% of product in the form of white powder with m.p. 226–230 °C; ${\left[\alpha \right]}_{27}^{D}$ = −16° (c = 1, DMF). ^1^H-NMR (DMSO-*d*_6_) δ, ppm: 0.82–0.88 (6H, m, 2C^δ^H_3_ Leu), 1.45 (2H, t, *J* = 14.25 Hz and *J* = 7.26 Hz, C^β^H Leu), 1.56–1.73 (1 H, m, C^γ^H_2_ Leu), 2.95 and 3.09 (2 H, 2dd, C^β^H Trp), 3.39 (2H, s, CH_2_C_6_H_5_), 4.15 (1H, dd, *J* = 15.27 Hz and *J* = 7.73 Hz C^α^H Leu), 4.62 (1H, m, C^α^H Trp), 6.97–7.38 (10H, m, Aryl), 7.07 and 7.58 (2 H, 2s, NH_2_ amide), 7.77 (1H, d, *J* = 8.38 Hz, NH Leu), 8.28 (1H, d, *J* = 7.92 Hz, NH Trp), 10.81 (1H, s, NH indole).

*CH_3_(CH_2_)_4_C(O)-l-Trp-l-Leu–NH_2_ (GD-107).* The 0.65 g (1.92 mmol) of H–l-Trp–l-Leu–NH_2_ was dissolved in 50 mL of DMF, then 0.45 g (2.11 mmol, 10% excess) of CH_3_(CH_2_)_4_C(O)Su was added and stirred for 12 h at room temperature. The solvent was evaporated to half the volume. The residue was diluted with 70 mL of ethylacetate and washed twice with 70 mL 5% NaHCO_3_, then with 70 mL 3% H_2_SO_4_ and then with 70 mL of distilled water. The organics were dried over anhydrous MgSO_4_ for half an hour; the solvent was filtered and evaporated. The yield was 0.74 g (92%), white powder with a m.p. 166 °C, ${\left[\alpha \right]}_{27}^{D}$ = −20° (c = 1, DMF). ^1^H-NMR (DMSO-*d*_6_) δ, ppm: 0.82–0,88 (9H, m, 2C^δ^H_3_ Leu and CH_3_ chain), 1.32 (2H, m, C^δ^H_2_), 1,45 (2H, m, C^β^H Leu), 1,56–1,73 (1H, m, C^γ^H_2_ Leu), 1.62 (4H, m, C^γ^H_2_C^β^H_2_), 2.63–2.68 (2H, m, C^α^H_2_), 2.95 and 3.09 (2 H, m, C^β^H Trp), 3.39 (2H, s, CH_2_C_6_H_5_), 4.15 (1H, m, C^α^H Leu), 4,62 (1 H, m, C^α^H Trp), 6.97–7.38 (10H, m, Aryl), 7.07 and 7.58 (2H, 2d, NH_2_ amide), 7.77 (1H, d, *J* = 7.73 Hz, NH Leu), 8.28 (1H, d, *J* = 8.Hz, NH Trp), 10.81 (1H, s, NH indole).

*N-Ph(CH_2_)_2_C(O)-d-Trp-l-Leu–NH_2_ (GD-128).* Obtained similarly to *N*-Ph(CH_2_)_2_C (O)-l-Trp-l-Leu–NH_2_ from 1.00 g (3.16 mmol) of H-d-Trp-l-Leu–NH_2_ and 1.17 g (4.74 mmol, 15% excess) of Ph(CH_2_)_2_C(O)Su. The yield was 1.04 g (74%) of the product in the form of a cream powder, with a m.p. 122 °C, ${\left[\alpha \right]}_{26}^{D}$ = −5° (c = 1, DMF). ^1^H-NMR (DMSO-*d*_6_, δ, ppm): 0.82–0,88 (6H, m, 2C^δ^H_3_ Leu), 1.45 (2H, t, C^β^H Leu), 1.56–1.73 (1H, m, C^γ^H_2_ Leu), 2.38 (2H, m, CH_2_C_6_H_5_), 2.69 (2H, m, CH_2_CO), 2.90–3.09 (2 H, m, C^β^H Trp), 4.22 (1H, dd, C^α^H Leu), 4.54 (1H, m, C^α^H Trp), 6.99–7.20 (10H, m, Aryl), 7.33 and 7.60 (2H, 2 s, NH_2_ amide), 7.93 (1H, d, *J* = 7.92 Hz, NH Leu), 8.08 (1H, d, *J* = 8.56 Hz, NH Trp), 10.80 (1H, s, NH indole).

*N-Ph(CH_2_)_2_C(O)-l-Trp-d-Leu–NH_2_ (GD-123)***.** Obtained similarly to *N*-Ph(CH_2_)_2_C (O)-l-Trp-l-Leu–NH_2_ from 1.00 g (3.16 mmol) of H-l-Trp-d-Leu–NH_2_ and 1.17 g (4.74 mmol, 15% excess) of Ph(CH_2_)_2_C(O)Su. The yield was 1.12 g (79%) of the product in the form of a cream powder, with a m.p. 210 °C (with decomposition), ${\left[\alpha \right]}_{26}^{D}$ = +17.7° (c = 1, DMF). ^1^H-NMR (DMSO-*d*_6_, δ, ppm): 0.78–0,81 (6H, m, 2C^δ^H_3_ Leu), 1.35 (2 H, m, C^β^H Leu), 1.60–1.74 (1H, m, C^γ^H_2_ Leu), 2.35–2.41 (2H, m, CH_2_C_6_H_5_), 2.69–2.74 (2H, t, *J* = 7.82 Hz, CH_2_CO), 2.91–3.03 (2H, m, C^β^H Trp), 4.22 (1H, dd, *J* = 8.29 Hz and *J* = 4.47 Hz, C^α^H Leu), 4.54 (1H, m, C^α^H Trp), 6.99–7.20 (10 H, m, Aryl), 7.33 and 7.60 (2H, 2 s, NH_2_ amide), 7.93 (1H, d, *J* = 7.73 Hz, NH Leu), 8.08 (1H, d, *J* = 8.47 Hz, NH Trp), 10.80 (1H, s, NH indole). 

*N-Ph(CH_2_)_2_C(O)-Gly-l-Leu–NH_2_ (GD-129).* The mixture of 2.51 g (10.29 mmol) TFA*[l-LeuNH_2_], 2.00 mL (11.31 mmol) DIPEA and 30 mL DMF was stirred for 0.5 h. Then the solution of 3.39 g (11.31 mmol 10% excess) of Ph(CH_2_)_2_C(O)-Gly–OSu in 30 mL of DMF was added. This mixture was stirred for 12 h at room temperature. The solvent was evaporated to half the volume. The residue was diluted with 100 mL of water to obtain an insoluble product. The resulting precipitate was maintained for 24 h at +8 °C, washed with 100 mL of water and 100 mL of hexane. The yield was 1.50 g (46%) of product in the form of white powder with m.p. 162 °C, ${\left[\alpha \right]}_{29}^{D}$ = −5.2° (c = 1, DMF). ^1^H-NMR (DMSO-*d*_6_, δ, ppm): 0.85 (6H, m, 2C^δ^H_3_ Leu), 1.45 (2H, m, C^β^H Leu), 1.56–1.73 (1H, m, C^γ^H_2_ Leu), 2.37 (2H, t, CH_2_CO), 2.69 (2H, t, CH_2_C_6_H_5_), 3.56 (2H, m, CH_2_ Gly), 4.47 (1 H, dd, *J* = 8.29 Hz and *J* = 4.47 Hz, C^α^H Leu), 6.99–7.20 (10H, m, Aryl), 7.33 and 7.60 (2H, 2s, NH_2_ amide), 7.93 (1H, d, *J* = 7.36 Hz, NH Leu), 8.22 (1H, t, *J* = 5.59 Hz, NH Gly), 10.79 (1H, s, NH indole).

*N-Ph(CH_2_)_2_C(O)-l-Trp-Gly-NH_2_ (GD-125).* A total of 4.50g (14.3 mmol) of H-l-Trp-l-Gly-NH_2_ was dissolved in 50 mL of DMF, then 3.86 g (15.6 mmol, 10% excess) of Ph(CH_2_)_2_C(O)Su was added and stirred for 12 h at room temperature. The solvent was evaporated to half the volume. The residue was diluted with 100 mL of water to obtain an insoluble product. The resulting precipitate was maintained for 24 h at +8˚C, filtered, washed with 100 mL of water and with 50 mL of hexane. The yield was 4.49 g (70%) of product in the form of white powder with m.p. 118 °C, ${\left[\alpha \right]}_{29}^{D}$= −33° (c = 1, DMF). ^1^H-NMR (DMSO-*d*_6_, δ, ppm): 2.92–3.11 (2H, m, C^β^H Trp), 2.37 (2H, m, CH_2_CO), 2.69 (2H, t, CH_2_C_6_H_5_), 3.40 (2H, m, CH_2_ Gly), 4.34 (1 H, m, C^α^H Trp), 6.52–7.77 (10H, m, Aryl), 7.57 (1H, d, *J* = 7.73 Hz, NH Trp), 7.41 and 7.10 (2H, 2s, NH_2_ amide), 8.14 (1H, d, *J* = 7.36, NH Gly), 10.12 (1H, s, NH indole).

### 3.4. Pharmacology

#### 3.4.1. Animals

Adult male Balb/c mice and ICR mice weighing 19–25 g were used in this study. The animals were obtained from the nursery of laboratory animals of Shemyakin-Ovchinnikov Institute of Bioorganic Chemistry of the Russian Academy of Sciences and adapted for 2 weeks prior to testing in the Zakusov Institute of Pharmacology vivarium. The mice were maintained in an environmentally controlled facility with a 12 h light-dark cycle (lights on at 08:00 o’clock), the temperature of 22 °C ± 2 °C and the humidity of 60 ± 10%. All the mice were put on a standard diet with food and water available ad libitum. The animals were split into groups of 8 according to body weight with a deviation from the mean value of no more than ±10%. All animals were kept for 24 h in the experimental room at “home” cages before experiments. 

#### 3.4.2. Ethical Approval

The study complied with the requirements of Order of the Ministry of Health of the Russian Federation Number 199 “On Approval of the Rules of Good Laboratory Practice” and with accordance with GOST 33215-2014 “Guidelines for accommodation and care of animals. Environment, housing and management” (http://protect.gost.ru/document.aspx?control=7&id=202494) and Directive 2010/63/EU of the European Parliament and of the Council of 22 September 2010 “On the Protection of Animals Used for Scientific Purposes.” The animals were kept in accordance with the sanitary and epidemiological rules 2.2.1.3218-14 “On Sanitary and Epidemiological Requirements for the Design, Equipment, and Maintenance of Experimental Biological Clinics (Vivariums),” approved by the Resolution of the Chief State Sanitary Doctor of the Russian Federation Number 51 of 29 August 2014. All manipulations with the animals were approved by the Bioethical Commission. All the experiments were approved by the Institutional Animal Care and Use Committee of V.V. Zakusov Research Institute of Pharmacology, Moscow (order number 68 of 17 October 2016). The work was performed within the framework of the government contracts of the Ministry of Science and Higher Education of the Russian Federation (Project 0521-2019-0002) in accordance with the protocol number 28 from 5 December 2018 and protocol number 2 from 9 February 2019. The article doesn’t include human studies and studies using human biomaterials, and therefore doesn’t contradict the rules of the WMA Declaration of Helsinki.

#### 3.4.3. Open Field Test

The open field apparatus was constructed as a white plywood arena with a diameter of 75 cm with white boards of 30 cm high. The arena was illuminated with 1500 lx by shadowless lamps fixed at the height of 60 cm above the floor. The arena of the open field was divided by 4 concentric circles, which in turn were divided into sectors by radii in a way that the peripheral circle consisted of 16 identical curvilinear squares. All surfaces of the OP were smooth, matte, without protrusions. The markings on the floor of the OP were applied with black matte paint, which did not protrude above the floor’s surface.

The experiment was carried out in accordance with the procedure described in [25]. Adult male Balb/c mice were used in this experiment. After 30 min of intraperitoneal administration of vehicle and compounds in the form of suspension in 1% Tween 80 water solution, the animals were placed in a dark chamber for 1 min, and then on the one of the peripheral squares of the arena. The number of squares crossed in the periphery (SCP) and in the central regions (SCC), the number of entries into the center (C) and number of rearing (R) were traced during 3 min. After each individual test session, the equipment was washed with soap and water, cleaned with 70% ethanol and dried. Then, the subsequent mouse was tested.

#### 3.4.4. Elevated Plus-Maze Test (EPM)

The EPM apparatus consisted of four arms elevated by 40 cm above the floor, with each arm positioned at 90° relative to the adjacent arms. Two of the arms were enclosed with high walls (65 × 5 × 15 cm), and the other arms were connected via a central area (5 × 5 cm) to form a plus sign. The maze floor and the walls of enclosed arms were painted in black. The room was illuminated with 50 lx.

The experiment was carried out in accordance with the procedure described in [28]. Adult male ICR mice were used in this experiment. The animals were treated with distilled water and compounds as a suspension in 1% Tween 80 water intraperitoneally, 30 min prior to the test. The experiment was performed between 09:00 and 14:00. Each mouse was individually placed on the central platform facing toward an open arm. The frequency and durations of entries into the open and closed arms were observed for 5 min. An entry was counted when all four paws of the mouse entered an open or closed arm. The percentage of time spent (duration) in the open arms (100 × open/(open + enclosed)) and the percentage of the number of open arm entries (frequency, 100 × open/total entries) were calculated for each animal. The apparatus was thoroughly cleaned after each run. 

#### 3.4.5. Influence of the PK11195 on Anxiolytic Activity of GD-102 in EPM Test

The experiment was carried out in accordance with the procedure described in [28]. Adult male ICR mice were used in this experiment. The animals were divided into four groups, 8 mice in each group. Group 1 consisted of control animals. Control group received distilled water. Distilled water was administered twice: 60 min and 30 min before testing. Group 2: PK11195 (10.0 mg/kg) was administered 60 min before and distilled water 30 min before testing. Group 3: distilled water was administered 60 min before and compound GD-102 (0.5 mg/kg) 30 min before testing. Group 4: PK11195 (10.0 mg/kg) was administered 60 min before and compound GD-102 (0.5 mg/kg) 30 min before testing. Animal behavior was evaluated in EPM test. The animals were treated with vehicle and compounds as suspensions in 1% Tween 80 water solution intraperitoneally.

#### 3.4.6. Registration of Spontaneous Motor Activity

The infrared actimeter Panlab (Spain) was used to register spontaneous locomotor activity [33]. The system consists of 2 dimensional (x and y coordinates) square frames, a supporting frame and a control device. Each frame includes 16 × 16 infrared beams. The frames were controlled by a control unit and registered with the ActiTrack software, which exports the data via RS 232 serial port to the Microsoft Excel compatible format. Measured parameters: general motor activity (total indicator of horizontal activity and stereotypy), stereotyping, i.e., movement in the absence of movement, horizontal activity, maximum travel speed (cm/s), average travel speed (cm/s), distance travelled (cm). The animals were preliminarily adapted in the experimental chamber for 20 min to exclude the orientation component of motor activity. The measurements for spontaneous locomotor were carried out within 10 min.

#### 3.4.7. Statistical Analysis

The statistical analysis was carried out using a nonparametric analogue of analysis of variance according to Kruskal–Wallis with further processing according to the Mann–Whitney test with Bonferoni’s correction.

#### 3.4.8. Docking Simulations Details

Molecular docking for compounds was performed with the active center 2MGY in complex with PK1195 taken from the (RCSB Protein Data Bank) PDB database. Protein and peptides were prepared using the Schrodinger Protein Preparation Wizard using the protocol [34,35]. Water molecules were removed because they do not participate in the ligand–receptor interaction. Ligands were prepared by the Ligprep module using Epik in the Schrödinger package to enhance protonation and tautomeric states at 7.0 ± 2.0 pH units. Confgen with OPLS 2005 force field was used to generate conformers. Docking settings: flexible ligands, solid receptor. The binding site was defined as a box surrounding PK 11195 at a distance of 20 A. Docking was performed in Glide v8.1. using SP peptide settings, poses were rendered in Maestro.

Pisa program (“Protein interfaces, surfaces and assemblies” PISA service at the European Bioinformatics Institute web server (http://www.ebi.ac.uk/pdbe/prot_int/pistart.html)) was used for calculations of the ligand–receptor interfaces and buried surface areas [22]. The superpositions of the TSPO and ligands complexes were performed by Cα atoms of all the residues (the Lsqkab program from the CCP4 library was used). Results of modeling were visualized with PyMol software (The PyMOL Molecular Graphics System, Version 0.99 Schrödinger, LLC., New York, NY, USA).

## 4. Conclusions

We have created a new highly active dipeptide TSPO ligand, compound GD-102, using the original drug-based peptide design (DBPD) strategy. The obtained ligand exhibited anxiolytic activity in two behavioral tests in mice at a dose range of 0.01–0.5 mg kg i.p. The ligand properties regarding TSPO were confirmed by molecular docking, which showed the high affinity of GD-102 structure to the active site of the receptor, and with pharmacological analysis using a specific TSPO antagonist, PK11195. Preliminary administration of PK11195 completely cancelled out the anxiolytic effect of GD-102; the animals’ behavior parameters remained at the control level. The study of the relationship between structure and activity showed the importance of the natural l-configuration of amino acid residues in the structure of dipeptide ligand for anxiolytic activity manifestation.

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
