# Peer review of "The New Dipeptide TSPO Ligands: Design, Synthesis and Structure–Anxiolytic Activity Relationship [Author-notes fn1-molecules-25-05132]"

_molecules, 2020, doi:10.3390/molecules25215132_

Round 1
Reviewer 1 Report
The authors are presenting the design, synthesis and structure activity relationship with dipeptide TSPO ligands. Although the workflow of the article is fluent a found at the end that there is no “new” TSPO ligand studied, sicnce all new compounds are less active compared to theGD-23 and GD102. Meanwhile I found in the literature the following article “Synthesis and Structure—Activity (Anxiolytic) Relationship Analysis of Leucyltryptophan Ligands of 18-KDA Translocator Protein Deeva; Pantileev; Rebeko; Rybina; Yarkova; Gudasheva; Seredenin [Pharmaceutical Chemistry Journal, 2020]” which is shown as part of this project and it is not mentioned inside the manuscript.
I would like from the authrors to revise the whole manuscript to convince the audience what is novel in the current project-manuscript.
Meanwhile I have some more comments:
Affiliation: Please state your country
Abstract:
line 19 The study of the relationship between structure and activity - > should be replaced by “structure activity relationship”
line 37: Examples of known TPSO lignads need to be shown as scheme
line 50-52 very long sentence and need to be rephrased
line 52 This strategy called drug-based 53 peptide design (DBPD) . There is no verb in sentence.
line 53-54 due to is found twice. Please rephrase.
Figure 1 should be scheme 1
Line 92: complexes are not near the global minimum. You should state how low was global minimum structure (kcal/mol) and compare the Glide Score of all new compounds.
Since you used Glide Software why not using Maestro for visualization?
Line 97: Docking data are divided to docking energy (glidescore) and molecular interactions. The authors have given details on the latter but they should also state the binding energy of the known inhibitor, compared to the new synthetic molucules.
The authors should explaine why the used the NMR structure of TSPO, and not the crystal structure one (pdb id: 4UC2, 4UC3, 4UC1)
Table 2 To complicate to follow. I would suggest a column-bar representation and the whole date to be transferred to supplementary material together with experimental data.
Line 213 conFiguration: what it means? Please correct it.
Table 3: Titles are cut. Please fix width of each column or decrease font size
Experimental
You need to provide details for docking simulations.
Figure 3,4,5. Non polar hydrogens should be undisplayed. Font size of labels should be increased.
Please rephrase: “TSPO contact residues are shown by wire Figures” to “TSPO contact residues are shown with wire representation. Please remove the label “no license file – for evaluation only. (0 days remaining).
Table 1 should be replaced with Ligand interaction diagram. Those figure can be reproduced easily from Maestro software. Since you used Glide you also have access to Maestro software.
Author Response
Dear Reviewer! We are sending You our response to Your Comments:
Point 1: The authors are presenting the design, synthesis and structure activity relationship with dipeptide TSPO ligands.
Response 1. The article is not devoted to all TSPO dipeptide ligands, but only to tryptophanilleucines. Other articles describe tryptophanilisoleucines (GD-23) [Gudasheva, T.A., Deeva, O.A., Mokrov, .GV., Dyabina, A.S., Yarkova, M.A., Seredenin, S.B. Design, Synthesis and anxiolytic activity evaluation of N-acyl-tryptophanyl-containing dipeptides, potential TSPO ligands. Medicinal Chemistry 2019, 15 (4), 383-399] and leucyltryptophanes. [Deeva, O. A., Pantileev, A. S., Rebeko, A. G., Rybina, I. V., Yarkova, M. A., Gudasheva, T. A., & Seredenin, S. B. (2020). Synthesis and structure — activity (Anxiolytic) relationship analysis of leucyltryptophan ligands of 18-kDa translocator protein. Pharmaceutical Chemistry Journal, 1-11].
Point 2: Although the workflow of the article is fluent a found at the end that there is no “new” TSPO ligand studied, since all new compounds are less active compared to the GD-23 and GD-102.
Response 2. The new highly active TSPO ligand in this article is compound GD-102. Previously it was briefly described only as academic thesis, as noted in the reference on the title page. Publication in the form of academic thesis is not an obstacle to the full publication of the current manuscript. Compound GD-23 is used for comparison.
Point 3. Meanwhile I found in the literature the following article “Synthesis and Structure—Activity (Anxiolytic) Relationship Analysis of Leucyltryptophan Ligands of 18-KDA Translocator Protein Deeva; Pantileev; Rebeko; Rybina; Yarkova; Gudasheva; Seredenin [Pharmaceutical Chemistry Journal, 2020]” which is shown as part of this project and it is not mentioned inside the manuscript.
Response 3. The found article [Synthesis and structure-anxiolytic activity relationship analysis of leucyltryptophan ligands of 18 kDa translocator protein, Deeva, Pantileev, Rebeko, Rybina, Yarkova, Gudasheva, Seredenin, Pharmaceutical Chemistry Journal, 2020] is not a part of the current manuscript. It is devoted to another group of TSPO dipeptide ligands, with the amino acid sequence Leu-Trp, but not Trp-Leu, as presented in the current manuscript. The article describing the development of the leucyl-tryptophan-containing ligands is the most recent work and it had not been published at the time of submission of the current manuscript. The reference is now provided because the publisher's imprint data of this article has appeared.
Point 4. I would like from the authrors to revise the whole manuscript to convince the audience what is novel in the current project-manuscript.
Response 4. In this work all dipeptides are new, except GD-23, which was given for comparison. The preparation of GD-102, its direct analogs and the structure-anxiolytic activity relationship among them are presented in this manuscript for the first time.
Thus, synthesis of GD-102 and its direct analogs and studying their anxiolytic activity is a new work; publication in academic thesis is not an obstacle to full publication.
Point 5: Affiliation: Please state your country
Response 5: We have stated our country in affiliation: “Baltiyskaya, 8, Moscow, 125315, Russian Federation”
Point 6: Abstract: line 19 The study of the relationship between structure and activity - > should be replaced by “structure activity relationship”
Response 6. “The study of the relationship between structure and activity” is replaced with “structure activity relationship”
Point 7. line 37: Examples of known TPSO ligands need to be shown as scheme
Response 7: We have represented the structures of known TSPO ligands at the scheme.
Point 8: line 50-52 very long sentence and need to be rephrased
Response 8: Rephrased: «An original strategy for creation highly active, non-toxic, physiological drugs based on dipeptides, which are structurally similar to non-peptide biologically active compounds was developed in Zakusov Research Institute of Pharmacology [12, 13]. This strategy called drug-based peptide design (DBPD). Due to their low molecular weight and the presence of an active transport system, dipeptides are able to penetrate biological barriers. Due to the small number of bonds that are targets of proteases, they are enzymatically stable up to the possibility of oral administration»
to
«An original strategy for creation highly active, non-toxic, physiological drugs based on dipeptides was developed in Zakusov Research Institute of Pharmacology [12, 13]. The strategy consists in constructing such dipeptide that are structurally similar to non-peptide biologically active compounds. This approach was called drug-based peptide design (DBPD). Due to their low molecular weight and the presence of an active transport system, dipeptides are able to penetrate biological barriers. A small number of bonds that are targets of proteases provide their enzymatic stability up to the possibility of oral administration.»
Point 9: line 52-53 This strategy called drug-based peptide design (DBPD) . There is no verb in sentence.
Response 9: see response 8
Point 10 : line 53-54. due to is found twice. Please rephrase
Response 10: see response 8
Point 11: Figure 1 should be scheme 1
Response 11: "Figure 1" is changed to "Scheme 1",
Point 12: Line 92: complexes are not near the global minimum. You should state how low was global minimum structure (kcal/mol) and compare the Glide Score of all new compounds.
Response 12: We agree with you. In the revised manuscript, we provide the data of Glide score and Glide emodel in table 1 to show how low was global minimum of the complexes (kcal/mol).
Point 13: Since you used Glide Software why not using Maestro for visualization?
Response 13: We preferred PyMol for the visualization. But in accordance with your comments we also performed visualization in Maestro. The material will be added to the supplement.
Point 14: Line 97. Docking data are divided to docking energy (glide score) and molecular interactions. The authors have given details on the latter but they should also state the binding energy of the known inhibitor. compared to the new synthetic molucules.
Response 14: Here are given the Glide Score and Glide emodel of docked complexes for all new synthesized compounds in comparison with alpidem and known inhibitor PK11195. The data will be placed in the supplement.
|
Ligand |
Glide gscore |
Glide emodel. kcal/mol |
|
Alpidem |
-8.89 |
-66.73 |
|
PK-11195 |
-8.08 |
-59.16 |
|
GD-102 |
-9.60 |
-71.97 |
|
GD-107 |
-8.75 |
-57.08 |
|
GD-108 |
-8.77 |
-44.05 |
|
GD-123 |
-9.03 |
-65.85 |
|
GD-125 |
-8.69 |
-69.04 |
|
GD-128 |
-9.15 |
-51.67 |
|
GD-129 |
-8.06 |
-66.39 |
|
GD-23 |
-8.76 |
-52.85 |
Point 15: The authors should explain why they used the NMR structure of TSPO. and not the crystal structure one (pdb id: 4UC2. 4UC3. 4UC1).
Response 15: The NMR structure of TSPO (PDB ID 2MGY) was used in the current study. as it is the structure of the mammalian (mouse) TSPO. Also. this structure is in the complex with the classical ligand. compound PK11195. docked into the active cavity. At the same time. the crystal structures 4UC2. 4UC3. 4UC1 given in the PDB are the microorganisms’ TSPO structures and are presented without ligands.
Point 16: Table 2 Too complicate to follow. I would suggest a column-bar representation and the whole date to be transferred to supplementary material together with experimental data.
Response 16: We have presented the full open field screening data in a form of table to better understanding.
Point 17: Line 213 conFiguration: what it means? Please correct it
Response 17: A typographic error occurred while editing the uploading draft-manuscript. It is the configuration at the C-alpha asymmetric carbon atom in the amino acid moiety.
Point 15: Table 3: Titles are cut. Please fix width of each column or decrease font size
Response 15: The cuttings in the words of the headings was apparently due to a mismatch in the A4 sheet format after editing the manuscript file. We fixed the size of the column in the table.
Point :16 Experimental. You need to provide details for docking simulations.
Response 16: We have added details for docking simulations to the experimental section (see point 4.4.8.)
Point 17: Figure 3.4.5. Non polar hydrogens should be undisplayed. Font size of labels should be increased.
Response17 : We have fixed the Figures in the revised manuscript.
Point 18: Please rephrase: “TSPO contact residues are shown by wire Figures” to “TSPO contact residues are shown with wire representation. Please remove the label “no license file – for evaluation only. (0 days remaining).
Response 18: We have fixed the figures headlines and removed the label “no license file - for evaluation only. (0 days remaining)
Point 19: Table 1 should be replaced with Ligand interaction diagram. Those figures can be reproduced easily from Maestro software. Since you used Glide you also have access to Maestro software.
Response 19: We have replaced table 1 with the ligand interaction diagrams, reprodused in the Maestro software. In the revised manuscript in is represented Figure 2 a,b,c.

Reviewer 2 Report
The present manuscript describes the design and synthesis of a series of novel dipeptide TSPO ligands with the subsequent investigation of their structure-anxiolytic activity relationship. The analysis of anxiolytic-like behaviours in mice subjected to a range of doses of synthesised compounds in the open field test allowed the authors to select the most promising anxiolytic TSPO ligand, which was dipeptide GD-102. Anxiolytic properties of GD-102 were then confirmed in the elevated plus-maze test. Moreover, the blockade of TSPO by PK11195, a specific TSPO antagonist, abolished GD-102-induced anxiolytic phenotype in mice, confirming that anxiolytic properties of GD-102 are TSPO-dependent but also that TSPO might serve as a potential therapeutic target in the treatment of anxiety.
The paper meets the urgent need for developing new pharmacotherapies for anxiety-related disorders. The chemical part of this study, in terms of both methods and data analysis, seems solid and does not give rise to any objections. However, at least few strong objections come up with the behavioural part of the presented research and additional clarifications need to be provided. Specific comments are given below:
Data analysis and presentation:
In general, I have great concerns about this part of the paper. See my comments below.
- The authors mention in section 4.4.6. that Kruskal-Wallis test was used. However, this is the only place where this test ever appears.
- The results from the OF test should be analysed using one-way ANOVA or its nonparametric equivalent (when appropriate) and followed by the post hoc test. The P- and F-value should be then given, at least in the supplementary material. The same is for the results included in Table 3.
- The statistical analysis for the results presented in Table 4 should be performed using two-way ANOVA, because with no doubts, in this case, there are two independent variables (two different treatments: I treatment - РК11195/vehicle, and II treatment – GD-102/vehicle).
- I took a closer look at the results in Table 2 and to my surprise, something is quite not right with the controls. I focused on SCP and it varies from 4.63 through ~16.00 to 35.88. Why the results for the control groups are so divergent? Moreover, SCP values for GD-123, GD-128, GD-118, GD-125 and GD-129 are equal the same, meaning 35.88 (9.03). So for all ligands mentioned above, there was one the same control group. Meanwhile, GD-116, GD-119, GD-107 and GD-108 have their own control group. Separate controls were also for GD-23 and GD-102. That is very confusing. I am not exactly sure how the experiments were exactly planned and performed.
- Table 2 could stay as it is, as it is more like the presentation of the results from a screening study. However, the results from the EPM test should be presented in figures as the percentages of open arms entries and the percentages of the time spent in the open arms, as well as the total number of arm entries, which may serve as the indicator of locomotor activity. Unfortunately, the way the EPM results are presented in the tables is nothing but beating about the bush.
OF test:
All due respect but the authors cited the paper of one of the co-authors of this manuscript dated in 1979 (!!!). And to be honest not much one can find there regarding the detailed description of the method.
- Why did you decide to use the TLA score as the indicator of anxiolysis instead of time spent or distance travelled in the centre zone of the OF? You must be aware that changes in locomotor activity will affect the results of the OF test. Maybe all you have observed was just hyperlocomotion and not anxiolysis? Discuss this issue and provide additional references.
- The size of the box (diameter = 1 m) is at least twice bigger than usually used and recommended. Additionally, the observation time was only 3 minutes. I am rather sceptical about the chosen procedure parameters.
- “The arena was illuminated by four 75 W shadowless lamps…” This information is not very useful. Please define light conditions in lux.
- What is “vehicle”? Is it just distilled water or distilled water with Tween? It must be clarified when what kind of solutions was injected as controls in the manuscript throughout.
EPM test:
- Please define light conditions in lux.
- Why did you decide to place mice in the EPM facing one of the open arms instead of one of the closed arms?
- What detergents were used to clean the apparatus?
- The headings in sections 4.4.3. and 4.4.4. should be rewritten in the manner like in section 4.4.5.
References:
The references need updating. 14 out of 32 references were published before 2000! I was curious about how many articles about TSPO were published in 2020 so far. PubMed finds 160 results when already restricted to the command TSPO[Title/Abstarc]. Of course, it does not beg the question but surely speaks volumes.
Other comments:
Line 37: “models[3,4]” – missing space
Line 37: “As examples of the TSPO ligands, are compound…” – Rephrase this sentence. It is a bit “clumsy”.
РК11195
Line 47: “… and in a number of other behavioral tests,” – missing references
Line 48: “... passed clinical trials or IS used in clinical practice.” – missing verb
Line 146: “The superposition of alpidem and GD-23 conformations docked in Table 23..”- In Table 23?
Lines 52-53: “This strategy IS called drug-based peptide design (DBPD).” – missing verb
Line 559: “… animals were approved by the Bioethical Commission.” – Please give the number of the Ethics Committee approval.
Table 3 and 4: word splitting in table headers
Author Response
Dear Reviewer! We are sending You our comments:
Data analysis and presentation:
Point 1. The authors mention in section 4.4.6. that Kruskal-Wallis test was used. However, this is the only place where this test ever appears. The results from the OF test should be analysed using one-way ANOVA or its nonparametric equivalent (when appropriate) and followed by the post hoc test. The P- and F-value should be then given, at least in the supplementary material. The same is for the results included in Table 3. The statistical analysis for the results presented in Table 4 should be performed using two-way ANOVA, because with no doubts, in this case, there are two independent variables (two different treatments: I treatment - РК11195/vehicle, and II treatment – GD-102/vehicle).
Response 1. Statistical analysis of the data in Tables 2-4 was carried out by a nonparametric method, since among the entire amount of data there are parameters with a distribution that is different from normal. We used nonparametric methods everywhere for uniformity.
The statistical analysis in this article was carried out using a nonparametric analogue of analysis of variance according to Kruskal-Wallis with further processing according to Mann-Whitney test with Bonferoni's correction.
In a revised manuscript we presented the data in the form of “Me (q25; q75)” in the tables 2, 3, 4.
Point 2. I took a closer look at the results in Table 2 and to my surprise, something is quite not right with the controls. I focused on SCP and it varies from 4.63 through ~16.00 to 35.88. Why the results for the control groups are so divergent? Moreover, SCP values for GD-123, GD-128, GD-118, GD-125 and GD-129 are equal the same, meaning 35.88 (9.03). So for all ligands mentioned above, there was one the same control group. Meanwhile, GD-116, GD-119, GD-107 and GD-108 have their own control group. Separate controls were also for GD-23 and GD-102. That is very confusing. I am not exactly sure how the experiments were exactly planned and performed.
Response 2. The activity studies of different compounds in mice were conducted in different months and different years. This is due to the fact that the compounds were synthesized at different times. Protocol # 06 from 16 Sept 2015; Protocol # 28 from 05 Dec 2018; Protocol # 02 from 09 Feb 2019.
The experiments for small doses of 0.01-0.05 mg/kg and for large doses of 0.1-5 mg/kg of GD-102 were also performed at different times and therefore had different control groups.
Point 3. Table 2 could stay as it is, as it is more like the presentation of the results from a screening study. However, the results from the EPM test should be presented in figures as the percentages of open arms entries and the percentages of the time spent in the open arms, as well as the total number of arm entries, which may serve as the indicator of locomotor activity. Unfortunately, the way the EPM results are presented in the tables is nothing but beating about the bush.
Response 3: We presented the results of the EPM test in figures; the tables with the data from EPM test were transferred to the supplement.
OF test:
Point 4:. All due respect but the authors cited the paper of one of the co-authors of this manuscript dated in 1979 (!!!). And to be honest not much one can find there regarding the detailed description of the method.
Response 4: We agree that in reference [Seredenin, S.B.; Vedernikov, A.A. Effect of psychotropic drugs on behavior of inbred mice under emotional stress. Bull. Exp. Biol. Med. 1979, 88 (1), 714-716] the method is not described in details, so we replaced it with a primary source reference, which describes the illuminated open field method in Balb/c mice thoroughly [Borodin, P.M.; Shuler, L .; Beliaev, D.K .; Genetic stress problems. The genetic analysis of mice behavior in stress conditions. Genetics 1976; 12, 12: 62-71]
Point 5: Why did you decide to use the TLA score as the indicator of anxiolysis instead of time spent or distance travelled in the centre zone of the OF? You must be aware that changes in locomotor activity will affect the results of the OF test. Maybe all you have observed was just hyperlocomotion and not anxiolysis? Discuss this issue and provide additional references
Response 5: Yes, we agree that the main parameter for the anxiolysis measurement is increasing of the central activity of animals. Since we use the mice with freeze-to-stress response, we evaluated anxiolytic activity in terms of the compound's effect on central activity so and motor activity (see reference [Borodin, P.M.; Shuler, L.; Beliaev, D.K.; Genetic stress problems. The genetic analysis of mice behavior in stress conditions. Genetics, 1976, 12, 12, p. 62-71]). We used increasing of motor activity to estimate the presence of anxiolytic activity. In addition, after the OF test, the animals were tested for spontaneous locomotor activity using the infrared actimeter Panlab (Spain). All compounds in the studied dose range did not change the locomotor activity of mice compared to the control group. We have added the description of the protocol for measuring spontaneous locomotor activity in the experimental part.
Point 6. The size of the box (diameter = 1 m) is at least twice bigger than usually used and recommended. Additionally, the observation time was only 3 minutes. I am rather sceptical about the chosen procedure parameters
Response 6: Please, excuse us for description the rat open field apparatus. We have provided the correct settings for the open field apparatus for mice.
Point 7: “The arena was illuminated by four 75 W shadowless lamps…” This information is not very useful. Please define light conditions in lux.
Response 7: we have defined the light conditions in lux. The light conditions in OF were 1500 lx.
Point 8: What is “vehicle”? Is it just distilled water or distilled water with Tween? It must be clarified when what kind of solutions was injected as controls in the manuscript throughout.
Response 8: In the revised version of manuscript we have clarified that the control groups were injected with the solvent (1% Tween-80 solution in distilled water), which was used to dissolve the test compounds.
EPM test:
Point 9: Please define light conditions in lux.
Response 9: we have defined the light conditions in lux. The light conditions in EPM were 50 lx.
Point 10: Why did you decide to place mice in the EPM facing one of the open arms instead of one of the closed arms?
Response 10: This type of mice placement in the EPM is the additional stress factor.
Point 11: What detergents were used to clean the apparatus?
Response 11: The apparatus was cleaned with 70% alcohol and dried (described in the experimental).
References
Point 12. The references need updating. 14 out of 32 references were published before 2000! I was curious about how many articles about TSPO were published in 2020 so far. PubMed finds 160 results when already restricted to the command TSPO[Title/Abstract]. Of course, it does not beg the question but surely speaks volumes
Response 12: Yes, we agree that we have provided some references prior to 2000. Some references we have replaced or supplemented with newer ones. However, not all references can be replaced. We can not change the references associated with the original source of the description of behavioral and chemical techniques, as well as references indicating the physical and chemical constants of intermediate compounds.
Other comments:
Point 13: Line 37: “models[3,4]” – missing space
Response 13: Fixed.
Point 14: Line 37: “As examples of the TSPO ligands, are compound…” – Rephrase this sentence. It is a bit “clumsy”.
Response 14: Rephrased “As examples of the TSPO ligands, are compound ...” to “Among the most well-known TSPO ligands are:”
Point 15: Line 47: “… and in a number of other behavioral tests,” – missing references
Response 15: Fixed
Point 16: Line 48: “... passed clinical trials or IS used in clinical practice.” – missing verb
Response 16: Rephrased to “passed clinical trials or is used in clinical practice”
Point 17: Line 146: “The superposition of alpidem and GD-23 conformations docked in Table 23..”- In Table 23?
Response17: A typographic error occurred after editing the uploaded manuscript file. Fixed "Table 23" to "TSPO".
Point 18: Lines 52-53: “This strategy IS called drug-based peptide design (DBPD).” – missing verb
Response 18: Fixed from “This strategy called drug-based peptide design (DBPD).” to “This strategy was called drug-based peptide design (DBPD).”
Point 20: Line 559: “… animals were approved by the Bioethical Commission.” – Please give the number of the Ethics Committee approval.
Response 20: We have given the Ethics Committee Approvals’ numbers.
Point 21: Table 3 and 4: word splitting in table headers
Response 21: The splitting of words in the table headers occurred due to a mismatch in the A4 sheet format after editing the manuscript file.

Round 2
Reviewer 1 Report
The authors have answered to all my questions and revised the manuscrpit as proposed.
I agree to accept the manuscript in the current form.
Author Response
Dear Reviewer!
Thank You for Your attention to our work, review of our manuscript and helpful comments.
Kind Regards, Authors
Reviewer 2 Report
Additional comments to the authors:
Response 1. Have you analyzed the results presented in Fig. 7 with two-way ANOVA as you were asked?
Response 2. This response does not explain why the results for the control groups vary that much, i.e. from 4.63 to 35.88, as previously presented (?!) I understand that the outcome for the control groups differ depending on the time points when the experiments were conducted. But such a huge difference? It might suggest highly unrepeatable experimental conditions. Hence, comparing different ligands between different protocols has no sense at all. You should emphasize and comment on this clearly obvious study limitation.
Response 10. Please provide additional comment in the text with a supporting reference.
Author Response
Point 1. Have you analyzed the results presented in Fig. 7 with two-way ANOVA as you were asked?
Response 1. We have analyzed the results presented in Fig. 7 (ex-table 4) with two-way ANOVA as we were asked.
Two-way ANOVA results with Tukey's multiple comparisons test.
- Time spent in open arms, s
|
Pairwise comparisons |
p values |
|
GD-102 vs H2O |
0,0032 |
|
PK11195 vs H2O |
0,9953 |
|
GD-102+PK11195 vs H2O |
0,9970 |
|
GD-102 vs PK11195 |
0,0017 |
|
GD-102 vs PK11195+GD-102 |
0,0053 |
|
PK11195 vs PK11195+GD-102 |
0,9711 |
- Time spent in center, s
|
Pairwise comparisons |
p values |
|
GD-102 vs H2O |
0,9849 |
|
PK11195 vs H2O |
0,9977 |
|
GD-102+PK11195 vs H2O |
0,9358 |
|
GD-102 vs PK11195 |
0,9983 |
|
GD-102 vs PK11195+GD-102 |
0,7860 |
|
PK11195 vs PK11195+GD-102 |
0,8679 |
- Time spent in closed arms, s
|
Pairwise comparisons |
p values |
|
GD-102 vs H2O |
<0,0001 |
|
PK11195 vs H2O |
0,9999 |
|
GD-102+PK11195 vs H2O |
0,9641 |
|
GD-102 vs PK11195 |
<0,0001 |
|
GD-102 vs PK11195+GD-102 |
<0,0001 |
|
PK11195 vs PK11195+GD-102 |
0,9469 |
- Number of entries in open arms, n
|
Pairwise comparisons |
p values |
|
GD-102 vs H2O |
0,0052 |
|
PK11195 vs H2O |
0,9935 |
|
GD-102+PK11195 vs H2O |
0,9122 |
|
GD-102 vs PK11195 |
0,0100 |
|
GD-102 vs PK11195+GD-102 |
0,0258 |
|
PK11195 vs PK11195+GD-102 |
0,9788 |
- Number of entries in closed arms, n
|
Pairwise comparisons |
p values |
|
GD-102 vs H2O |
0,7143 |
|
PK11195 vs H2O |
0,6565 |
|
GD-102+PK11195 vs H2O |
0,6565 |
|
GD-102 vs PK11195 |
0,1418 |
|
GD-102 vs PK11195+GD-102 |
0,1418 |
|
PK11195 vs PK11195+GD-102 |
>0,9999 |
- Time spent in open arms, %
|
Pairwise comparisons |
p values |
|
GD-102 vs H2O |
<0,0001 |
|
PK11195 vs H2O |
>0,9999 |
|
GD-102+PK11195 vs H2O |
0,9966 |
|
GD-102 vs PK11195 |
<0,0001 |
|
GD-102 vs PK11195+GD-102 |
<0,0001 |
|
PK11195 vs PK11195+GD-102 |
0,9974 |
- Time spent in center, %
|
Pairwise comparisons |
p values |
|
GD-102 vs H2O |
0,9843 |
|
PK11195 vs H2O |
0,9975 |
|
GD-102+PK11195 vs H2O |
0,9386 |
|
GD-102 vs PK11195 |
0,9984 |
|
GD-102 vs PK11195+GD-102 |
0,7886 |
|
PK11195 vs PK11195+GD-102 |
0,8691 |
- Time spent in closed arms, %
|
Pairwise comparisons |
p values |
|
GD-102 vs H2O |
<0,0001 |
|
PK11195 vs H2O |
0,9999 |
|
GD-102+PK11195 vs H2O |
0,9642 |
|
GD-102 vs PK11195 |
<0,0001 |
|
GD-102 vs PK11195+GD-102 |
<0,0001 |
|
PK11195 vs PK11195+GD-102 |
0,9465 |
- Total numbers of entries in the arms, n
|
Pairwise comparisons |
p values |
|
GD-102 vs H2O |
0,5085 |
|
PK11195 vs H2O |
0,6595 |
|
GD-102+PK11195 vs H2O |
0,5085 |
|
GD-102 vs PK11195 |
0,9946 |
|
GD-102 vs PK11195+GD-102 |
>0,9999 |
|
PK11195 vs PK11195+GD-102 |
0,9946 |
- Relative time spent in open arms , topen/topen+tclosed, %
|
Pairwise comparisons |
p value |
|
GD-102 vs H2O |
<0,0001 |
|
PK11195 vs H2O |
>0,9999 |
|
GD-102+PK11195 vs H2O |
0,9950 |
|
GD-102 vs PK11195 |
<0,0001 |
|
GD-102 vs PK11195+GD-102 |
<0,0001 |
|
PK11195 vs PK11195+GD-102 |
0,9978 |
- Relative number of entries in open arms , nopen/nopen+nclosed, %
|
Pairwise comparisons |
p values |
|
GD-102 vs H2O |
0,0032 |
|
PK11195 vs H2O |
0,9953 |
|
GD-102+PK11195 vs H2O |
0,9970 |
|
GD-102 vs PK11195 |
0,0017 |
|
GD-102 vs PK11195+GD-102 |
0,0053 |
|
PK11195 vs PK11195+GD-102 |
0,9711 |
Figures in the attached file represent the screenshots from GraphPad Prism software.
Point 2. This response does not explain why the results for the control groups vary that much, i.e. from 4.63 to 35.88, as previously presented (?!) I understand that the outcome for the control groups differ depending on the time points when the experiments were conducted. But such a huge difference? It might suggest highly unrepeatable experimental conditions. Hence, comparing different ligands between different protocols has no sense at all. You should emphasize and comment on this clearly obvious study limitation
Response 2. Experiments of the anxiolytic activity of GD-102 in the dose range of 0.1-5.0 mg/kg in the Open Field test were carried out in the autumn of 2015. The resulting low control data may be associated with weather disturbances at this time. Therefore, we decided to remove this data and the corresponding Protocol No.6 from 16 Sept 2015. This does not break the logic of the article.
Point 10. Please provide additional comment in the text with a supporting reference.
Response 10. Different authors follow varied protocols.
Deacon [Deacon, R. M. (2013). The successive alleys test of anxiety in mice and rats. JoVE (Journal of Visualized Experiments), (76), 1-7], indicated that mouse was placed at the closed end of closed arm facing the wall.
In the article Fraga et al. [Fraga, D. B., Olescowicz, G., Moretti, M., Siteneski, A., Tavares, M. K., Azevedo, D., Rodrigues, A. L. S. (2018). Anxiolytic effects of ascorbic acid and ketamine in mice. Journal of Psychiatric Research, 100, 16-23.] reported, that the mouse was individually placed in the center of the maze facing between closed arm and open arm.
- Naderi et al. [Naderi, N., Haghparast, A., Saber-Tehrani, A., Rezaii, N., Alizadeh, A. M., Khani, A., Motamedi, F. (2008). Interaction between cannabinoid compounds and diazepam on anxiety-like behaviour of mice. Pharmacology Biochemistry and Behavior, 89(1), 64-75.] states, that each animal was placed in the central platform facing one of the open arms.
Lister [Lister, R. G. (1987). The use of a plus-maze to measure anxiety in the mouse. Psychopharmacology, 92(2), 180-185.] reported, that each mouse was placed in the center of the plus maze facing to the open arms.
Rico et al. [Rico, J. L., Muñoz-Tabares, L. F., Lamprea, M. R., & Hurtado-Parrado, C. (2019). Diazepam Reduces Escape and Increases Closed-Arms Exploration in Gerbils After 5 min in the Elevated Plus-Maze. Frontiers in psychology, 10, 748] reported, that animals were placed in the central area of the EPM facing one of the open arms.
Rodgers el al. [Rodgers, R., Lee, C., & Shepherd, J. K. (1992). Effects of diazepam on behavioural and antinociceptive responses to the elevated plus-maze in male mice depend upon treatment regimen and prior maze experience. Psychopharmacology, 106(1), 102-110] reported, that animals were individually placed onto the central platform facing an open arm.
Our research group has followed to the protocol in which the mouse is placed facing the open arm. We run all experiments in the identically way. This technique has been validated using classic tranquillizer diazepam, for example in the article Mokrov et al. [Mokrov, G. V., Deeva, O. A., Gudasheva, T. A., Yarkov, S. A., Yarkova, M. A., & Seredenin, S. B. (2015). Design, synthesis and anxiolytic-like activity of 1-arylpyrrolo [1, 2-a] pyrazine-3-carboxamides. Bioorganic & medicinal chemistry, 23(13), 3368-3378.].
